# Solid-Phase Extraction of Catechins from Green Tea with Deep Eutectic Solvent Immobilized Magnetic Molybdenum Disulfide Molecularly Imprinted Polymer

**DOI:** 10.3390/molecules25020280

**Published:** 2020-01-09

**Authors:** Wanwan Ma, Kyung Ho Row

**Affiliations:** Department of Chemistry and Chemical Engineering, Inha University, Incheon 402-701, Korea; wanwanma@inha.edu

**Keywords:** molecular imprinted polymers, magnetic solid-phase extraction, deep eutectic solvents, molybdenum disulfide, catechins

## Abstract

A type of molecular-imprinted polymer with magnetic molybdenum disulfide as a base and deep eutectic solvent as a functional monomer (Fe_3_O_4_@MoS_2_@DES-MIP) was prepared with surface molecular imprinting method. It was applied as the adsorbent for the selective recognition and separation of (+)-catechin, (−)-epicatechin, (−)-epigallocatechin, (−)-epicatechin gallate, and (−)-epigallocatechin gallate in green tea in the process of magnetic solid-phase extraction (MSPE) combined with high-performance liquid chromatography (HPLC). The structure of Fe_3_O_4_@MoS_2_@DES-MIP was characterized by Fourier transform infrared spectroscopy and field emission scanning electron microscopy. The adsorption properties and selective recognition ability on (−)-epigallocatechin gallate and the other four structural analogues were examined and compared. The results show that the polymer has excellent selective recognition ability for (−)-epigallocatechin gallate, and its adsorption capacity was much higher than that of structural analogues. The Fe_3_O_4_@MoS_2_@DES-MIP not only has the special recognition ability to template a molecule, but also can be separated by magnets with high separation efficiency and can be used in MSPE.

## 1. Introduction 

Green tea is considered a healthy drink and has become one of the most widely consumed beverages [1]. In addition, it has been used in the food, pharmaceutical, chemical, and light industries for thousands of years, especially in Asia, because of its numerous benefits, such as cancer prevention, inhibition of oxidation, and lowering blood pressure [2,3,4]. Catechins are the unique components of green tea that endow green tea with its physiological effects [5,6]. (+)-Catechin (C), (−)-epicatechin (EC), (−)-epigallocatechin (EGC), (−)-epicatechin gallate (ECG), and (−)-epigallocatechin gallate (EGCG) are the main catechins in green tea. Their excellent anti-oxidation, anti-cancer, and anti-cancer activities have prompted research into the separation and purification of those bioactive compounds [7,8,9]. The separation and purification technologies include distillation, freezing clarification, organic solvent extraction, and ion exchange. High purity catechins can be obtained through a combination of the above technologies [10]. On the other hand, these techniques do not meet the requirements of the separation and purification of trace analogs in complex systems, and these analogs are often difficult to analyze.

Molecular imprinting technology (MIT) is a type of molecular specific recognition technology that has developed rapidly in recent years [11]. The preparation technology of molecularly imprinted polymers (MIP) can be divided into two categories: embedding method [12] and surface molecularly imprinting method [13,14]. Encapsulation includes mainly bulk polymerization, suspension polymerization, emulsion polymerization, and precipitation polymerization. On the other hand, with the development of molecularly imprinted technology, the molecularly imprinted polymers prepared by the encapsulation method have the disadvantages of a large particle size, few imprinted dots, and few binding sites [15,16,17]. The surface molecularly imprinted method can be divided into the sacrificial carrier method, polymerization plus membrane method, and chemical grafting method [18]. The molecularly imprinted polymer synthesized by this method has more effective imprinting points owing to its large specific surface area. Moreover, its imprinting layer is thinner than that prepared by the embedding method, with faster desorption and adsorption rates, and template leakage can be eliminated. Because of the above advantages, surface molecularly imprinted technology has attracted increasing attention. This technology has broad application prospects in drug separation, sensors, solid-phase extraction, and food safety detection.

Molybdenum disulfide (MoS_2_) is a two-dimensional layered material with a natural adjustable bandgap and its monolayers are bonded via weak van der Waals forces and exhibit strong covalent S-Mo-S bonds in the plane [19]. MoS_2_ is considered an appropriate candidate as an alternative to graphene owing to its similar structure, and it has begun to be applied to MIP processes [20,21]. Magnetic MoS_2_ by combining Fe_3_O_4_ with MoS_2_ improved the potential of this material as an absorbent for magnetic solid-phase extraction (MSPE) because it could be easily and directly collected with an extra magnet.

Deep eutectic solvents (DESs) are a new type of ionic liquid-like solvent that has attracted attention because of their excellent physical and chemical properties, such as better chemical stability, designability, and recyclability, than ionic liquids (ILs) [22,23]. Moreover, DESs can be produced using inexpensive raw materials and offer environmental protection and simple synthesis processes [24,25,26]. Therefore, DESs are considered green solvents, as important as the ionic liquids, to replace traditional organic solvents. According to previous research [27,28], several types of DES have been applied as the functional monomer in the synthesis of MIPs, whereas a DES with the composition of vinyl-pyrrolidone (VP) and malonic acid (MA), which exhibits biocompatibility and new functions, has rarely been applied and discussed [29].

In this paper, a magnetic surface molecularly imprinted polymer (MIP) was prepared using a surface molecular imprinting method with nanometer magnetic MoS_2_ as a carrier; EGCG, as the template molecule; DES consisting of VP and MA as the functional monomer; and glycol dimethacrylate as the cross-linking agent. The structure of the DES-based MIP (Fe_3_O_4_@MoS_2_@DES-MIPs) was characterized, and its extraction and selective recognition ability for EGCG is discussed.

## 2. Results and Discussion

### 2.1. Characterization of Fe_3_O_4_@MoS_2_@DES-MIP

The DES, Fe_3_O_4_@MoS_2_-MPS base and Fe_3_O_4_@MoS_2_@DES-MIP with DES functional monomer were analyzed by FTIR spectroscopy, and the FTIR spectra is given in Figure 1. MoS_2_ showed a very strong characteristic stretching vibration peak of Mo-S around 600 cm^−1^, which also can be observed from the spectrums of Fe_3_O_4_@MoS_2_-MPS and Fe_3_O_4_@MoS_2_@DES-MIP, which is evidence of the successful coasting of MoS_2_ at those two materials. Fe_3_O_4_@MoS_2_-MPS and Fe_3_O_4_@MoS_2_@DES-MIPs both have absorption peaks around 575 cm^−1^, which is the characteristic absorption peak of Fe-O in Fe_3_O_4_; it is speculated that the Fe_3_O_4_ formed on the surface of MoS_2_ polymer was successful_._ The peaks arounds 1200 cm^−1^ and 1500 cm^−1^ were assigned to the C-N stretching vibration and C-N bending vibration, indicating the successful preparation of DES from VP and MA. In addition, the strong broad peak around 3000 cm^−1^ of O-H vibration peak was attributed to a large number of hydrogen bonds formed in DES. These bands confirmed the presence of the DES and base of Fe_3_O_4_@MoS_2_ in the Fe_3_O_4_@MoS_2_@DES-MIP structure.

The FESEM (field emission scanning electron microscopy) images of MoS_2_, Fe_3_O_4_@MoS_2_, and Fe_3_O_4_@MoS_2_@DES-MIP are shown in Figure 2, and the structural characteristics of the polymer can be clearly seen. As we can see from the figure of MoS_2,_ the morphology of MoS_2_ was a schistose structure and the particle size distribution was uniform. It can be seen from Fe_3_O_4_@MoS_2_ that there was a thick crust of Fe_3_O_4_ wrapped on the surface of MoS_2_, indicating that Fe_3_O_4_@MoS_2_ has been successfully prepared. As shown in the part of Fe_3_O_4_@MoS_2_@DES-MIP, the thickness of the modified and imprinted layer is increased. Thus, the polymer formed by the polymerization reaction has been successfully grafted on the outer surface of the Fe_3_O_4_@MoS_2_, which indicates that the magnetic surface molecularly imprinted polymer has been successfully prepared. In addition, the more cavities on the polymer particles can lead to the increase of the adsorption capacity and the mass transfer rate to release and recombine with the analyte.

### 2.2. Adsorption Capacity

#### 2.2.1. Adsorption Isotherm

The static adsorption curves of Fe_3_O_4_@MoS_2_@DES-MIP and non-imprinted polymer (NIP) at different temperatures (20, 30, and 40 °C) are shown in Figure 3. It can be seen from the figure that its adsorption amount increases with increasing temperature. The reason for this may be that the MIP polymerizes at a higher temperature and has better adsorption capacity in a high temperature environment. Obviously, the equilibrium adsorption capacity of MIP is more than six times larger than that of NIP at the same temperature.

#### 2.2.2. Adsorption Kinetics

The adsorption kinetic curve of MIP at 40 °C is shown in Figure 4. It can be observed that the adsorption capacity for EGCG by MIP is sharply increased before 30 min. However, it eventually tends to the equilibrium condition around 40 min, because the adsorption process needs to slowly enter the pores and combine with the recognition site. While there is no specific adsorption in the adsorption process of NIP, the adsorption amount is relatively small, and it had reached equilibrium in about 20 min.

#### 2.2.3. Selective Adsorption Capacity

The selective adsorption capacity test of C, EC, EGC, ECG, and EGCG by MIP and NIP was prepared with EGCG as templates are shown in Figure 5. The result showed that when EGCG was applied as the template, the corresponding MIP exhibited excellent adsorption for its templated molecule of EGCG, with an adsorption capacity of 70.5 mg × g^−1^, while its adsorption capacities for C, EC, EGC, and ECG were greatly reduced, which are 25.5, 16.7, 16.9, and 20.9 mg × g^−1^; the recoveries of those four analytes are 31.8%, 13.5%, 21.1%, and 26.2%, respectively. Similarly, MIP using EC as a template showed excellent selectivity for EC, and the adsorption of the other four analytes was very small. In addition, there was no significant difference in the adsorption amount of NIP to each substance, indicating that MIP has a higher selective adsorption capacity for a template molecule. The imprinting factor of the MIP with EGCG as a template was calculated to be 9.23 when the initial concentration of EGCG was 2.00 mg × mL^−1^. According to the comparison of the before and after of the MSPE process for the mixture of the five analytes, a significant decrease in the concentration of EGCG was found, while the concentrations of the other four substances remained almost unchanged. Therefore, the prepared Fe_3_O_4_@MoS_2_@DES-MIP showed specific adsorption selectivity and EGCG can be identified well from structural analogs.

### 2.3. MIP Adsorption of EGCG in Green Tea

The chromatograms of the green tea extracts before and after Fe_3_O_4_@MoS_2_@DES-MIP adsorption are shown in Figure 6. By comparing the chromatograms before and after adsorption, it was found that the peak area of EGCG changed obviously. It was calculated that about 97.9% of EGCG had been absorbed by MIP, while the concentration of other substances in green tea remained basically unchanged. Therefore, the MIP obtained in the experiment could selectively identify EGCG from green tea extractant and could be widely used in solid phase extraction of EGCG. 

### 2.4. Validation and Applications

The Fe_3_O_4_@MoS_2_@DES-MIP-MSPE-HPLC method was evaluated, and the correlation coefficients (R^2^), limits of detection (LOD), and limits of quantification (LOQ) were obtained for the five catechins. Table 1 showed the calibration Curves, LOQs and LODs for the five analytes. Good linearities with R^2^ all above 0.990 were achieved, and the LODs from the five catechins were ranged from 0.36 to 1.20 mg/L. The accuracy and precision of the method were evaluated by performing three replicates of the fortified samples on the same and on different days, and the results are shown in Table 2. The relative standard deviations (RSD) for the intra-day and inter-day at the three levels of 10, 50, 100 µg × mL^−1^ were determined and all the RSD were less than 5.79%. The results showed that the reported method is reproducible and can be used for the analysis of the analytes in real samples. The repeatability of the prepared Fe_3_O_4_@MoS_2_@DES-MIP was tested; the adsorbent after the extraction of the target molecule from green tea was collected by an extra magnet and applied to the extraction processes five times. The result showed that the target molecule could be extracted out again but the recoveries for the extracted amount were decreased from 98% to 55%.

## 3. Materials and Methods 

### 3.1. Reagents and Materials

Vinyl pyrrolidone (VP), iron (III) chloride hexahydrate (FeCl_3_·6H_2_O), sodium acetate, malonic acid (MA), and molybdenum disulfide (MoS_2_) were purchased from Sigma Chemical Co. (St. Louis, MO, USA). Acetic acid and ethylene glycol were procured from Duskan Pure Chemical (Kyungki-do, Korea). Methanol and ethanol were acquired from Fisher Scientific (Seoul, Korea). Ammonium persulfate (APS), 3-methacryloxypropyltrimethoxysilane (MPS), Ethylene glycol dimethacrylate (EGDMA), and 2-methylpropionitrile (AIBN) were purchased from Daejung Chemicals & Metals (Gyonggido, Korea). The standard chemicals of (+)-catechin (C, ≥98%), (−)-epicatechin (EC, ≥90%), (−)-epigallocatechin (EGC, ≥95%), (−)-epicatechin gallate (ECG, ≥95%), and (−)-epigallocatechin gallate (EGCG, ≥98%) were supplied by Tokyo Chemical Industry (Tokyo, Japan). All other solvents used in the experiment were either of HPLC or analytical grade.

### 3.2. Instrumentation and Conditions

The chromatographic analysis was carried on a YL9100 HPLC System (Incheon, Korea) that consisted of three parts: a YL9110 quaternary pump, an injector with a 20 mL sample loop, and a UV-visible dual channel detector. The data acquisition system was effectuated with the Autochro-2000 software (Younglin, Korea). HPLC separation was carried out with the mobile phase of water/methanol/acetic acid (70:30:0.01, *v*/*v*/*v*) with a flow rate of 0.8 mL/min, and an OptimaPak C18 column (5 µm, 250 × 4.6 mm, id, RS tech Corporation, Daejeon, Korea) was applied to the stationary phase. The injection volume was 10 µL, the column temperature was kept at 30 °C, and UV-vis detector was set at a wavelength of 278 nm. 

The characterizations of the systemized materials were performed with a field emission scanning electron microscopy (FE-SEM S-4200, Hitachi, Ontario, Canada) and Fourier transform infrared spectra (FTIR, Vertex 80 V Bruker, Billerica, MA, USA). The morphology evaluation was examined by FE-SEM S-4200 with an acceleration voltage of 15 kV (pixel size: 0.5 nm) to observe the structure and size changes of the MIP. FTIR was performed over the range of 4000–400 cm^−1^, at a scan rate of 20 scans/min using a KBr pellet to analyze the graft bonding of the polymer surface at each stage of the experiment.

### 3.3. Preparation of DESs

The VP-based DESs were synthesized by the heating method according to Fu’s work [22]. Briefly, 0.1 mol VP and 0.1 mol MA were placed in a 100-mL round-bottom flask, the mixture was constantly stirred and kept in a water bath with the temperature set at 80 °C. The DES could be obtained until the mixture becomes evenly and clear. 

### 3.4. Preparation of Fe_3_O_4_@MoS_2_-MPS Nanoparticles

Fe_3_O_4_@MoS_2_ core-shell composites were prepared according to Chen’s method [21], but there was a slight modification. 12.9 g FeCl_3_·6H_2_O and 2.94 g sodium acetate were added into a beaker with 100 mL ethylene glycol, then 3.137 g MoS_2_ nanoparticles were dispersed in the mixture homogeneously with the help of ultrasound sonication for 2 h with a 20 min intermittent time interval. Following, the mixture was transferred to a 200-mL Teflon-lined stainless-steel autoclave and sealed to heat at 150 °C for 10 hours. After the mixture was cooled down to room temperature, the obtained black product was rinsed with distilled water several times with a centrifuge and the synthesized Fe_3_O_4_@MoS_2_ nanoparticles were collected by an extra magnet.

To prepare the Fe_3_O_4_@MoS_2_ nanoparticles modified with MPS, the obtained Fe_3_O_4_@MoS_2_ nanoparticles were dispersed in 50 mL of anhydrous toluene containing 5 mL of MPS, and the mixture was stirred for 8 h at 60 °C. The Fe_3_O_4_@MoS_2_ microspheres modified by MPS (Fe_3_O_4_@MoS_2_-MPS) were washed respectively with DMSO and ethanol several times, then collected by an extra magnet.

### 3.5. Preparation of Fe_3_O_4_@MoS_2_ @DES-MIP

The DES-MIPs were prepared via the surface imprinted polymerization method as follows: A 1 mg template of EGCG and 1 mmol of functional monomer of DES were added to the via and then they were dispersed in toluene (15 ml) by sonication, then the mixture stood at room temperature for 6 hours before use. Thirty milligrams of the prepared Fe_3_O_4_@MoS_2_-MPS, 2 mmol cross-linker of EGDMA, 10 mg initiator of AIBN, and 1 mL porogen of ethanol–water mixture with the volume ratio of 9:1 (*v*/*v*) were all placed together in a mixture and dissolved under ultrasonic agitation for 5 min. Next, the pre-assembly solution was dropped into the above solution under a N_2_ gas sweep, and then mechanically stirred at 60 °C overnight. After the polymerization was completed, the products were repeatedly washed with acetonitrile and deionized water until the supernatant was transparent, then methanol-acetic acid (volume ratio 9:1) was used to finally remove the template molecules. Finally, it was washed with methanol to neutrality and dried under vacuum at 50 °C to obtain the Fe_3_O_4_@MoS_2_@DES-MIP; the schematic illustration of the preparation of Fe_3_O_4_@ MoS_2_@DES-MIP is shown in Figure 7.

The same method was applied to the preparation of DES-based magnetic non-imprinted polymers (Fe_3_O_4_@ MoS_2_@DES-NIP) and magnetic imprinted polymers (Fe_3_O_4_@MoS_2_-MIP) without adding the template of EGCG and DES, respectively.

### 3.6. Sample Pretreatment

The standard products of the five kind of catechins were accurately weighed and the standard stock solutions of them were prepared with deionized water. The standard stock solution was diluted with deionized water to obtain standard solutions of 1, 5, 10, 50, 100, and 200 μg·× mL^−1^, respectively, then the standard curve of them was obtained with the help of HPLC.

After drying the Green tea in an oven (50 °C) and grinding it to a powder, 1 g powder of the green tea sample was weighed and soaked in 90 °C distilled water for 20 mins. After centrifugal separation, the suspension was then filtered to obtain the extraction samples; finally, the sample solution was filtered through a 0.45 μL nylon membrane before being applied further.

### 3.7. Absorption Capacity Test of Fe_3_O_4_@MoS_2_@DES-MIP

#### 3.7.1. Adsorption Thermodynamic Experiment

A series of EGCG aqueous solutions (0.5, 1.0, 1.5, 2.0, 2.5, and 3.0 mg × mL^−1^) were prepared, and 15 mg of MIP or NIP was added to the EGCG solution of 3.00 mL. They were then shaken at a constant temperature for 120 min at different temperatures (20, 30 and 40 °C). After fully adsorbed, the adsorbed solution was separated magnetically, and its peak area was determined by HPLC with the help of EGCG standard curve. The EGCG concentration in the solution after adsorption was measured three times in parallel and then determined. The adsorption amount of EGCG was calculated by the Formula (1).
(1)$$Q=\frac{\left(C_{0}-C_{v}\right)*V\;}{m}$$

In the formula, Q (mg × g^−1^) is the adsorption amount of the target substance by the polymer, $C_{0}$ (mg × mL^−1^) and $C_{v}$ (mg × mL^−1^) are the initial concentrations of the target substance and the concentration of the standard solution after shaking for 120 min, V (mL) is the Volume of the added standard solution, and m (g) is the mass of the adsorbent of Fe_3_O_4_@MoS_2_@DES-MIP.

#### 3.7.2. Adsorption Kinetics Experiment

One-hundred milligrams of Fe_3_O_4_@MoS_2_@DES-MIP was added into 10 mL of EGCG solution with the concentration of 2.0 mg mL^−1^ with consistent shaking at 40 °C, and then the samples were separated and collected at different times. The contraction in the solution after adsorption was determined by HPLC. The adsorption amount of EGCG was calculated by Formula (1).

#### 3.7.3. Selection Adsorption Experiments 

Fifteen milligrams of MIPs were added to 3.0 mL of C, EC, EGC, ECG, and EGCG solutions (all at the concentration of 2.0 mg mL^−1^) while constantly shaking at 30 °C for 120 min, respectively. After it was fully adsorbed, the supernatants were separated from the polymer by an external magnet and then discarded. The equilibrium adsorption capacity of Fe_3_O_4_@MoS_2_@DES-MIP for the five substances was calculated and measured in parallel three times. Repeat the above steps to determine the adsorption capacity of these five substances with the imprinted polymer using EC as a template. The equilibrium adsorption capacity is calculated by Formula (1); the recognition specificity of the MIP is evaluated using the imprinting factor (α); and the value of α is calculated by Formula (2)
(2)$$\alpha \;=\;\frac{Q_{MIP}}{Q_{NIP}}\;\times 100\%$$

In the formula, Q_MIP_ and Q_NIP_ are the adsorption amounts (mg/g) of GA to imprinted and non-imprinted polymers, respectively.

In addition, 3.0 mL of a mixed standard solution of C, EC, EGC, ECG, and EGCG (all with a concentration of 2.0 mg × mL^−1^) was shaken with the addition of 15 mg of MIP at 30 °C for 120 min, and then analyzed by HPLC. The concentration of each component in the solution before and after adsorption was evaluated to evaluate the difference in the selective adsorption abilities of MIP to these five structural analogs.

## 4. Conclusions

The magnetic surface molecularly imprinted polymer of Fe_3_O_4_@MoS_2_@DES-MIP was successfully synthesized with EGCG as a template, Fe_3_O_4_@MoS_2_ as a base, and a new type of DES as a functional monomer, and it was applied in the MSPE-HPLC method for the selective recognition and separation of five kind of catechins (C, EC, EGC, ECG, and EGCG) in green tea. The results show that the adsorption capacity of MIP was about six times higher than that of NIP. The saturated adsorption capacity of EGCG was significantly higher than that of the other four catechins, which indicates that MIP has a specific selective recognition of EGCG. It was found that MIP could adsorb more than 98% of EGCG in green tea extractant and could be applied in the MSPE process. The application of MoS_2_ combined with Fe_3_O_4_ and the application of green solvent of DES instead of the traditional functional monomer may provide further exploration in the preparation of MIP.

## Figures and Tables

**Figure 1 molecules-25-00280-f001:**
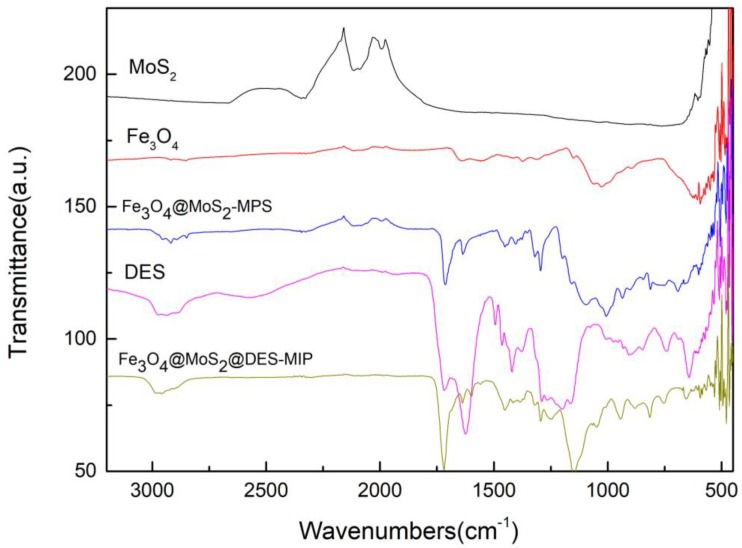
FTIR spectra of MoS_2_, Fe_3_O_4_, Fe_3_O_4_@ MoS_2_-MPS, DES, and Fe_3_O_4_@ MoS_2_@DES-MIP.

**Figure 2 molecules-25-00280-f002:**
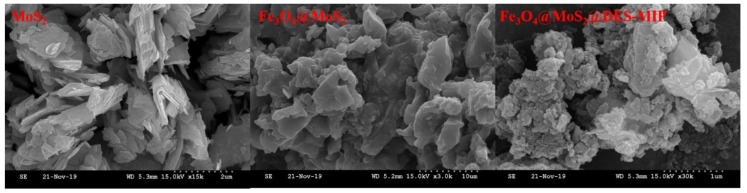
The FESEM images of the prepared MoS_2_, Fe_3_O_4_@MoS_2_, and Fe_3_O_4_@MoS_2_@DES-MIP.

**Figure 3 molecules-25-00280-f003:**
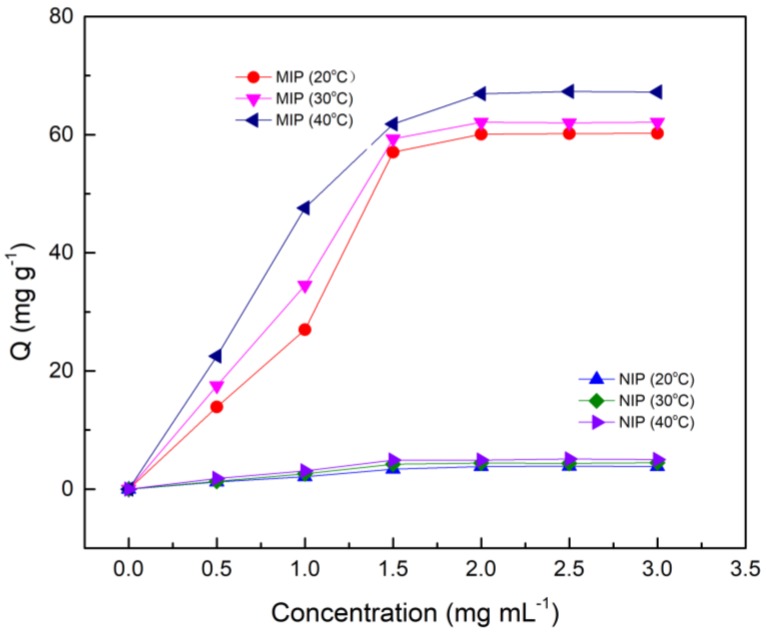
The static adsorption curves of Fe_3_O_4_@ MoS_2_@DES-MIP and Fe_3_O_4_@ MoS_2_@DES-NIP.

**Figure 4 molecules-25-00280-f004:**
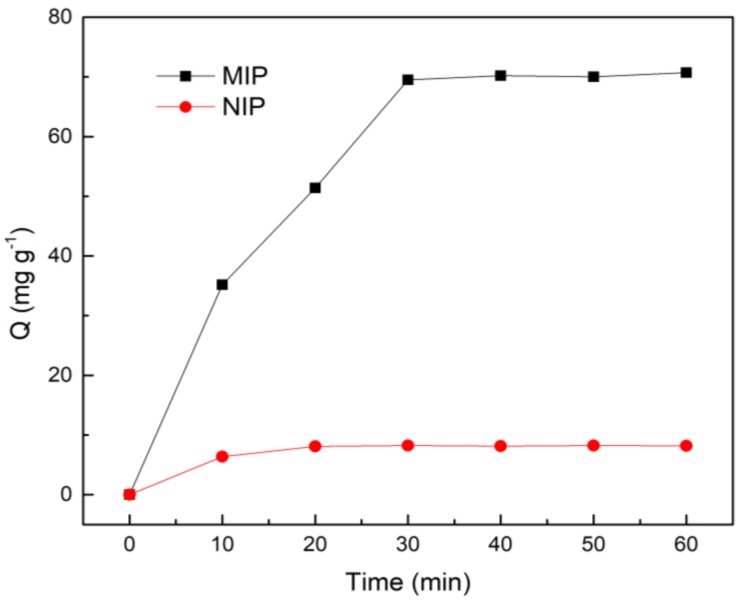
The adsorption kinetics curves of Fe_3_O_4_@ MoS_2_@DES-MIP and Fe_3_O_4_@ MoS_2_@DES-NIP.

**Figure 5 molecules-25-00280-f005:**
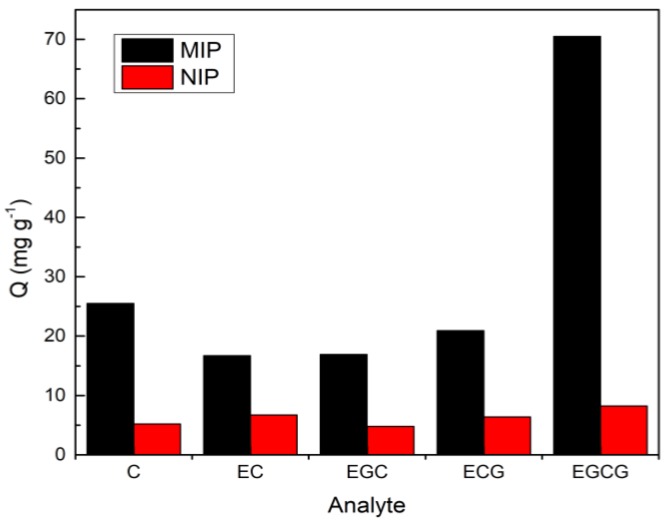
The selective adsorption capacity to the 5 catechins. Fe_3_O_4_@ MoS_2_@DES-MIP with EGCG template molecule and its corresponding NIP.

**Figure 6 molecules-25-00280-f006:**
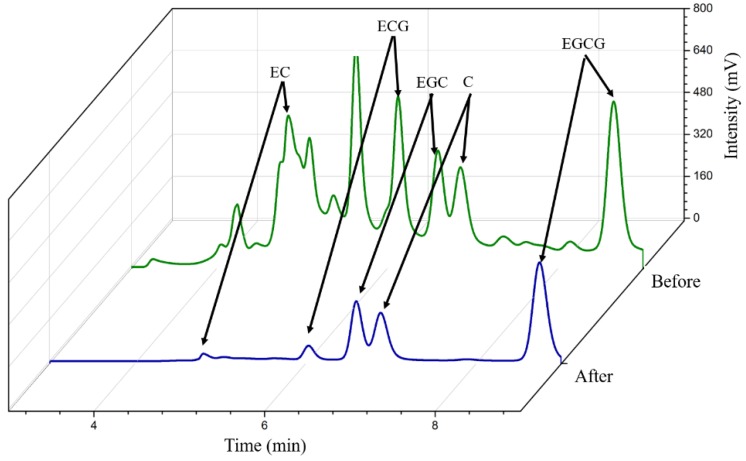
Chromatograms of green tea sample before and after MIP-SPE.

**Figure 7 molecules-25-00280-f007:**
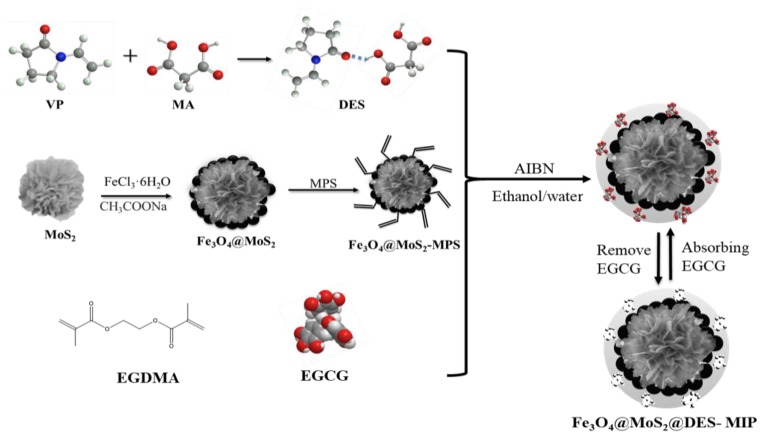
Schematic of the preparation of DES and Fe_3_O_4_@ MoS_2_@DES-MIP.

**Table 1 molecules-25-00280-t001:** Calibration Curves (*n* = 5), LODs, and LOQs for analytes.

Target	Equation	R2	LOQ (mg/L)	LOD (mg/L)
C	Y = 128.85x − 64.5	0.9992	0.60	0.20
EC	Y = 103.71x − 19.48	0.9991	1.20	0.40
EGC	Y = 467.2x − 14.61	0.9990	0.82	0.26
ECG	Y = 207.36x − 25.26	0.9990	0.36	0.10
EGCG	Y = 197.96x − 33.25	0.9994	0.50	0.15

**Table 2 molecules-25-00280-t002:** Magnetics solid-phase extraction method recoveries (*n* = 3) and relative standard deviation values of analytes.

Analyte	Concentration (µg/mL)	Intra-Day		Inter-Day	
		Recovery (%)	RSD (%)	Recovery (%)	RSD (%)
C	10	88.1	2.01	82.4	3.62
	50	85.4	3.15	77.2	2.28
	100	84.8	4.4	75.9	4.4
EC	10	84.2	1.91	78.4	2.96
	50	82.4	2.54	76.8	3.25
	100	80.7	5.79	74.7	4.01
EGC	10	84	2.09	76.2	2.85
	50	80.3	2.84	74.9	4.32
	100	79.5	4.22	73.3	5.48
ECG	10	84	2.09	76.2	2.85
	50	80.3	2.84	74.9	4.32
	100	79.5	4.22	73.3	5.48
EGCG	10	98.6	2.09	92.1	2.85
	50	95.7	2.84	88.9	4.32
	100	90.4	2.01	85.3	3.62

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
