# Peer review of "Solid-Phase Extraction of Catechins from Green Tea with Deep Eutectic Solvent Immobilized Magnetic Molybdenum Disulfide Molecularly Imprinted Polymer"

_molecules, 2020, doi:10.3390/molecules25020280_

Round 1

Reviewer 1 Report

The temperature and time experiments (figure 3 and 4) are not extremely relevant. Furthermore figures need more discussion; for instance for figure 6, there has not been explained what the different retention times are and which retention times stands for which target. 

In case you want to extract molecules from natural products with MIPs, it is benificial as well showing that the target molecule can be extracted out again and recovered

Author Response

Dear Editor,

Thank you very much for your approval and detailed comments for our manuscript: Manuscript ID molecules-677860, entitled " Solid-Phase Extraction of Catechins from Green Tea with Deep Eutectic Solvent Immobilized Magnetic Molybdenum Disulfide Molecularly Imprinted Polymer". According to your suggestions, we had checked the manuscript thoroughly and some parts of the manuscript were modified and corrected with "track changes" option.

Questions are answered term by term as follows:

Response to Reviewer 1 Comments

Point 1: The temperature and time experiments (figure 3 and 4) are not extremely relevant. Furthermore figures need more discussion; for instance for figure 6, there has not been explained what the different retention times are and which retention times stands for which target.

Response 1: Figure 3 showed the static adsorption curves which illustrated the influence of reaction temperature to adsorption capacity. Figure 4 showed the adsorption kinetics curves which illustrated the influence of reaction time to adsorption capacity. Static adsorption curves and adsorption kinetics curves were applied to set up the best adsorption conditions of temperature and time. And the more discussion for the figures were added in the manuscript. The retention times for different analytes were showed in the revised figure 6. Please check.

Point 2: In case you want to extract molecules from natural products with MIPs, it is benificial as well showing that the target molecule can be extracted out again and recovered

Response 2: Table 2 showed the recoveries of the magnetics solid-phase extraction method, and the prepared Fe3O4@MoS2@DES-MIP after the extraction of target molecule from green tea were collected by an extra magnet and applied to the extraction processes for 5 times, the result showed that the target molecule could be extracted out again but the recoveries for the extracted amount were decreased from 98% to 55%. The related part has been added in the manuscript. Please check.

Thank you!

Sincerely,

Prof. Kyung Ho Row
Department of Chemical Engineering, Inha University,

402-751, 253, Yonghyun-Dong, Nam-Gu, Incheon, Korea
Tel: 82-32-860-7470; Fax: 82-32-872-0959
E-mail: rowkho@inha.ac.kr

Reviewer 2 Report

Manuscript ID: molecules-677860

Title: Solid-Phase Extraction of Catechins from Green Tea with Deep Eutectic

Solvent Immobilized Magnetic Molybdenum Disulfide Molecularly Imprinted

Polymer

Authors: Kyung Ho Row *, Wanwan Ma

The present manuscript can be accepted for publication, provided the following items are considered:

Specific comments (page, line):

I suggest harmonizing all the numbers with a spacer after the measurement unit (e.g. page 3 line 81, 84, 86; page 8 line 186; page 11 line 255; page 9, line 210 and 211; page 10, line 237, 238 and 245 and so on) inside the whole manuscript. At page 1, line 15 insert a spacer between “chromatography” and “(HLPC)”. At page 2, line 43 a dot in missing. At page 2, line 60-61, the authors affirm that “it can be collected and easily directly using only a magnet”. I suggest correcting this sentence. At page 2, line 80 I suggest changing Fig. 1 in Figure 1. At pane 3, please correct the names of the axes of figure 1. At page 3, line 101 – 103, I suggest reformulating the sentence since it is not complete. At page 4, line 108 for the first time is described the acronyms NIP, please describe the acronyms. In figure 3 insert a spacer after the brackets in both the axes names. At page 4, line 117 – 122, many times are mentioned the terms “increase” and “adsorption”, making the sentence unpleased to be read. Please reformulate the sentence. In figure 4 insert a spacer after the brackets in both the axes names. Could be useful insert also at which temperature the experiments were conducted. At page 5, line 126 – 127, I suggest reformulating the sentence. At page 5, line 129 – 130, I suggest inserting, beside the quantity of compounds adsorbed, a percentage related to the difference adsorbtion capacity. At page 5, line 133 – 136, I suggest reformulating the sentence. At page 6, line 146, I suggest changing the words “changed a lot”. In figure 5 insert a spacer after the brackets “Q(“. I suggest defining and describe the picks in figure 6 and insert a spacer after the brackets in both the axes names. Moreover, I do not understand why the measure unit is in voltage since the detection typology is UV – VIS. Please clarify this point

Author Response

Dear Editor,

Thank you very much for your approval and detailed comments for our manuscript: Manuscript ID molecules-677860, entitled " Solid-Phase Extraction of Catechins from Green Tea with Deep Eutectic Solvent Immobilized Magnetic Molybdenum Disulfide Molecularly Imprinted Polymer". According to your suggestions, we had checked the manuscript thoroughly and some parts of the manuscript were modified and corrected with "track changes" option.

Questions are answered term by term as follows:

Response to Reviewer 2 Comments

Specific comments (page, line):

I suggest harmonizing all the numbers with a spacer after the measurement unit (e.g. page 3 line 81, 84, 86; page 8 line 186; page 11 line 255; page 9, line 210 and 211; page 10, line 237, 238 and 245 and so on) inside the whole manuscript. At page 1, line 15 insert a spacer between “chromatography” and “(HLPC)”. At page 2, line 43 a dot in missing. At page 2, line 60-61, the authors affirm that “it can be collected and easily directly using only a magnet”. I suggest correcting this sentence. At page 2, line 80 I suggest changing Fig. 1 in Figure 1. At pane 3, please correct the names of the axes of figure 1. At page 3, line 101 – 103, I suggest reformulating the sentence since it is not complete. At page 4, line 108 for the first time is described the acronyms NIP, please describe the acronyms. In figure 3 insert a spacer after the brackets in both the axes names. At page 4, line 117 – 122, many times are mentioned the terms “increase” and “adsorption”, making the sentence unpleased to be read. Please reformulate the sentence. In figure 4 insert a spacer after the brackets in both the axes names. Could be useful insert also at which temperature the experiments were conducted. At page 5, line 126 – 127, I suggest reformulating the sentence. At page 5, line 129 – 130, I suggest inserting, beside the quantity of compounds adsorbed, a percentage related to the difference adsorbtion capacity. At page 5, line 133 – 136, I suggest reformulating the sentence. At page 6, line 146, I suggest changing the words “changed a lot”. In figure 5 insert a spacer after the brackets “Q(“. I suggest defining and describe the picks in figure 6 and insert a spacer after the brackets in both the axes names. Moreover, I do not understand why the measure unit is in voltage since the detection typology is UV – VIS. Please clarify this point

Response:

According to your suggestion, all the numbers in the manuscript have been added with a spacer after the measurement unit. And the changes haven been e clearly highlighted using the "Track Changes" function.

At page 1, line 15 a spacer between “chromatography” and “(HLPC)” has been inserted.

At page 2, line 43 a dot between the two sentence has been added.

At page 2, line 60-61, the sentence has been modified and rewritten. “it could be easily and directly collected with an extra magnet”.

At page 2, line 80 the “Fig. 1” has been changed to “Figure 1”.

The name of the axes of figure 1 has been corrected and figure 1 has been replaced to the revised figure.

At page 3, line 101 – 103, the sentence has been modified and rewritten. “And the more cavities on the polymer particles can lead to the increase of the adsorption capacity and the mass transfer rate to release and recombine with the analyte.”

At page 4, line 108. The full name of non-imprinted polymer (NIP) has been added.

Figure 3 has been modified and the spacers after the brackets in both the axes names have been added, and figure 3 has been replaced by the revised one.

At page 4, line 117 – 122, the sentence has reformulated, the mentioned time of the terms “increase” and “adsorption” have been reduced. “It can be observed that the adsorption capacity for EGCG by MIP is sharply increased before 30 min.”

In figure 4, a spacer after the brackets in both the axes names have been inserted, and figure 4 has been replaced by the revised one.

The experiments were conducted at the temperature of 40℃ has been showed in the manuscript.

At page 5, line 126 – 127, the sentence has been reformulated. “The result showed that when EGCG was applied as the template, the corresponding MIP exhibited excellent adsorption for its’ templated molecular of EGCG, which with an adsorption capacity of 70.5 mg g-1

At page 5, line 129 – 130, the recovery percentage related to the difference adsorption capacity have been added.

At page 5, line 133 – 136, the sentence has been reformulated. “The imprinting factor of the MIP with EGCG as template was calculated to be 9.23 when the initial concentration of EGCG is 2.00 mg mL-1. According to the comparison of the before and after of MSPE process for the mixture of the five analytes, a significant decrease in the concentration of EGCG was found, while the concentrations of the other four substances remained almost unchanged.”

At page 6, line 146, the words “changed a lot” has been changed to “obviously changed”.

In figure 5, a spacer has been inserted after the brackets “Q(“, and figure 5 has been replaced by the revised one.

Figure 6 has been modified, the name of Y-axis has been corrected as “Intensity (mV)”, the spacer after the brackets in both the axes names have been added. And the revised figure 6 has been applied in the manuscript.

Thank you!

Sincerely,

Prof. Kyung Ho Row
Department of Chemical Engineering, Inha University,

402-751, 253, Yonghyun-Dong, Nam-Gu, Incheon, Korea
Tel: 82-32-860-7470; Fax: 82-32-872-0959
E-mail: rowkho@inha.ac.kr

Reviewer 3 Report

the paper is focused on the realisation and characterisation of a molecular-imprinted polymers with magnetic molybdenum disulfide as base and deep eutectic solvent as functional monomer  was prepared with surface molecularly imprinting method. It was applied as the adsorbent for the selective recognition and separation of some compounds in green tea in the process of magnetic solid-phase extraction combined with HPLC. the paper is well written and clear in each sections, the results obtained are well discussed.
my only concern is related principally on the reported state of art, the authors should better analyse the state of art of MIP in differents areas such as these works:
1)Soft-molecular imprinted electrospun scaffolds to mimic specific biological tissues. Biofabrication. 2018 Aug 20;10(4):045005.
2). SOFT-MI: a novel microfabrication technique integrating soft-lithography and molecular imprinting for tissue engineering applications. Biotechnol Bioeng. 2010 Aug 1;106(5):804-17. doi: 10.1002/bit.22740.
3) New biomedical devices with selective peptide recognition properties. Part 1: Characterization and cytotoxicity of molecularly imprinted polymers. J Cell Mol Med. 2007 Nov-Dec;11(6):1367-76. doi: 10.1111/j.1582-4934.2007.00102.x.

Author Response

Dear Editor,

Thank you very much for your approval and detailed comments for our manuscript: Manuscript ID molecules-677860, entitled " Solid-Phase Extraction of Catechins from Green Tea with Deep Eutectic Solvent Immobilized Magnetic Molybdenum Disulfide Molecularly Imprinted Polymer". According to your suggestions, we had checked the manuscript thoroughly and some parts of the manuscript were modified and corrected with "track changes" option.

Questions are answered term by term as follows:

Response to Reviewer 3 Comments

Comments and Suggestions for Authors

the paper is focused on the realisation and characterisation of a molecular-imprinted polymers with magnetic molybdenum disulfide as base and deep eutectic solvent as functional monomer  was prepared with surface molecularly imprinting method. It was applied as the adsorbent for the selective recognition and separation of some compounds in green tea in the process of magnetic solid-phase extraction combined with HPLC. the paper is well written and clear in each sections, the results obtained are well discussed.

my only concern is related principally on the reported state of art, the authors should better analyse the state of art of MIP in differents areas such as these works:

1)Soft-molecular imprinted electrospun scaffolds to mimic specific biological tissues. Biofabrication. 2018 Aug 20;10(4):045005.

2). SOFT-MI: a novel microfabrication technique integrating soft-lithography and molecular imprinting for tissue engineering applications. Biotechnol Bioeng. 2010 Aug 1;106(5):804-17. doi: 10.1002/bit.22740.

3) New biomedical devices with selective peptide recognition properties. Part 1: Characterization and cytotoxicity of molecularly imprinted polymers. J Cell Mol Med. 2007 Nov-Dec;11(6):1367-76. doi: 10.1111/j.1582-4934.2007.00102.x..

Response: Thanks for your recommend of those good papers which related to our work, and the literatures have been quoted in our revised manuscript.

Thank you!

Sincerely,

Prof. Kyung Ho Row
Department of Chemical Engineering, Inha University,

402-751, 253, Yonghyun-Dong, Nam-Gu, Incheon, Korea
Tel: 82-32-860-7470; Fax: 82-32-872-0959
E-mail: rowkho@inha.ac.kr

Round 2

Reviewer 1 Report

none anymore